

# Reduced fish diversity despite increased fish biomass in a Gulf of California Marine Protected Area

Georgina Ramírez-Ortiz[1], Héctor Reyes-Bonilla[2], Eduardo F. Balart[1], Damien Olivier[2,3], Leonardo Huato-Soberanis[1], Fiorenza Micheli[4] and Graham J. Edgar[5]

[1] Programa de Ecología Pesquera, Centro de Investigaciones Biológicas del Noroeste S.C., La Paz, Baja California Sur, México

[2] Departamento Académico de Ciencias Marinas y Costeras, Universidad Autónoma de Baja California Sur, La Paz, Baja California Sur, México

[3] Consejo Nacional de Ciencia y Tecnología, Ciudad de México, México

[4] Hopkins Marine Station and Center for Ocean Solutions, Stanford University, Pacific Grove, CA, United States of America

[5] Institute for Marine and Antarctic Studies, University of Tasmania, Hobart, Tasmania, Australia

Corresponding author
Georgina Ramírez-Ortiz,
gramirez@pg.cibnor.mx

## ABSTRACT

Multi-use marine protected areas (MUMPAs) are a commonly applied tool for marine conservation in developing countries, particularly where large no-take reserves are not socially or politically feasible. Although MUMPAs have produced benefits around the world, the persistence of moderate fishing pressure reduces the likelihood of achieving the primary objective of these areas, which is the conservation of ecosystems. In this study we used traditional and functional metrics to evaluate how fish assemblages changed through time in a MUMPA, including shifts in species responses and in ecological processes. We conducted visual censuses of fishes at Espíritu Santo Island, México (MUMPA; $N = 320$; 24°N, 110°W) from 2005 to 2017 to assess fish richness, size-distribution and density. Three functional indices were calculated using six traits (size, mobility, period of activity, aggregation, position in water column and diet): functional richness (volume occupied by species), dispersion (complementarity between species) and originality (inverse of functional redundancy). We compared fish diversity among three management zone types (sustainable fishing, traditional fishing and no-take zones), through a 13-year period, assessing which species increased or decreased in occurrence, density, and biomass, and how indices respond over time. Despite a general increase in biomass and stability in density and originality, we detected a reduction in fish biodiversity in the form of declines in species and functional richness, which could imply the risk of local extinction and decrease in certain ecosystem processes. In addition, changes in functional dispersion showed that some functions are losing representation through time. Although no single cause is apparent, such factors as competitive interactions, habitat loss and persistence of fishing pressure potentially explain these decreases. The rise in biomass was associated with a general increase in the average size, rather than increased biomass of commercial species, as the latter remained stable during the study period. Expansion of no-take areas, enforcement of fishing regulations, and surveillance in core zones, should be implemented to reverse the decline in particular species and to promote conservation of fish functional diversity in this MUMPA.

## INTRODUCTION

Marine Protected Areas (MPA) are the most common and promising tool for mitigating anthropogenic disturbance (mainly fishing pressure) on marine ecosystems (*Lester & Halpern, 2008*; *Lester et al., 2009*; *Bates et al., 2014a*). Nevertheless, different MPA schemes exist, from the banning of all fishing activities (marine reserves), to multi-use marine protected areas (MUMPA) that regulate fishing activities at different degrees, from restricted extraction (for area, species captured, and gear employed) to complete prohibition areas (*Agardy et al., 2003*; *Sala & Giakoumi, 2018*). The benefits of fully-protected marine reserves on fish species richness, density and biomass have been demonstrated in many studies throughout the world, while MUMPAs show controversial results (*Bates et al., 2014a*; *Coleman et al., 2015*; *Campbell et al., 2018*).

For example, one analysis of 20 MUMPAs exhibited positive but non-significant responses for species richness, density and biomass, because even though industrial fisheries are banned from these areas, a moderate fishing pressure persists (restricted mainly on the fishing gear allowed) and may affect the populations of target species (*Lester & Halpern, 2008*). In addition, some indirect effects of protection have been observed, especially in protected areas, where the maintenance or increase in the density of target species through reduced fishing pressure can cause a decline in non-target species due to competitive and predatory interactions (*Coleman et al., 2015*; *Ulate et al., 2018*). Considering that direct and indirect effects of protection can structure communities in opposing or uncorrelated directions (*Leenhardt et al., 2015*), it is necessary to evaluate temporal changes to understand ecosystem dynamics within MUMPAs, especially in those areas which aim to protect irreplaceable biodiversity.

The Gulf of California (GC) is considered a hotspot of biological diversity (*Roberts et al., 2002*) and a key region for the fishing industry, providing 70% of the total catch in México (*Cisneros-Mata, 2010*; *Díaz-Uribe et al., 2013*). Although human population density is relatively low in the region, it is currently growing rapidly, and the GC is not exempt from global coastal and marine degradation trends (*Lluch-Cota et al., 2007*; *Sagarin et al., 2008*; *Calderon-Aguilera et al., 2012*). Since the mid-1980s the Mexican government has established several MPAs to conserve biodiversity and control extraction of natural resources (*CONANP, 2007*). However, all the MPAs in the GC are MUMPAs with small no-take areas surrounded by "buffer" zones where fishing effort is restricted (*Rife et al., 2013*). This is a planned strategy because the enactment of large no-fishing marine reserves is not socially and politically feasible in many developing countries (*Halpern, 2003*), and thus MUMPAs are the preferred alternative to safeguard the regional biodiversity and ensure a variety of ecosystem services (poverty reduction, coastal protection, recreation, tourism, and carbon sequestration) in addition to fish stock enhancement (*Spalding et al., 2013*; *Caveen et al., 2014*).
Few long-term studies have assessed the effects of MPAs in the GC. A two year comparison of Parque Nacional Cabo Pulmo in 1999 and 2009 showed a rise in species richness and biomass, especially of large carnivore fish species (*Aburto-Oropeza et al., 2011*), but another study in the same MPA comparing two different years (1987 and 2003), revealed a decrease in general species richness and fish density (*Alvarez-Filip & Reyes-Bonilla, 2006*). Cabo Pulmo suffered from habitat deterioration between 1997 and 2003 due to impacts from hurricanes and El Niño Southern Oscillation events, which may have temporally damped the effect of protection. These observations underpin the limitations of short-term comparisons, especially those conducted between particular years, as they are not able to adequately capture the actual trends of natural and environmental variability, and the community response to them. On the other hand, a single continuous long-term study was carried out in the GC, at Parque Nacional Bahía de Loreto, another MUMPA (*Rife et al., 2013*). This 13-year long investigation revealed relatively stable total fish biomass but a decline in reported total fish landing, compared to open access sites. The authors concluded that, even if management strategies were not able to recover reef fish populations, at a minimum they should maintain the conditions of the ecosystem that existed when the park was established, which is not what they observed. However, a small no-take zone (less than 1% of the total area) in this MUMPA presented significant biomass increases associated to a greater abundance of herbivorous and zooplanktivorous fishes in this zone. This result emphasizes the need of creation of larger no-take areas to improve reef fish populations and ensure sustainable fisheries far into the future. Such long-term studies need to be repeated throughout the GC and worldwide, to provide robust general assessments of the value of MUMPAs in conserving the living heritage, and in supplying natural resources to human local communities, and guidance for improving their management.

Parque Nacional exclusivamente la zona marina del Archipiélago de Espíritu Santo (PNZMAES) was implemented in 2007 in the southwestern GC, a hotspot for reef fishes in the region (*Olivier et al., 2018*). This MUMPA encompasses exclusively the marine area around the entire Espíritu Santo archipelago and adjacent small isles, and is located less than 30 km from the major regional population center: the city of La Paz, the capital of the state of Baja California Sur, with over 300,000 inhabitants. The aim of this MPA is to conserve the ecosystems and ecological processes that occur in the area, while also allowing the use of the natural resources under an integrated management scheme (*CONANP-SEMARNAT, 2014*). Thanks to the collaboration of local fishermen, academics, civil organizations and federal authorities, PNZMAES is considered successful, and in 2018 became the first national park of México included in the Green List of Protected Areas by the International Union for the Conservation of Nature (IUCN). This is a relevant global accolade as the list includes only 46 areas recognized worldwide for effective management, governance, design and planning (*IUCN, 2018*).

The monitoring of reef ecosystems at PNZMAES began in 2005, two years before the declaration of the protected area in 2007, and has continued without interruption to date. Information derived from this systematic effort was used to design the management plan which was published in 2014, in which the no-take (1.4%) and ''buffer'' (98.6%)
zones were officially designated (*CONANP-SEMARNAT, 2014*). The period covered by the monitoring program provides an important opportunity to assess the effect of a MUMPA on the diversity of multiple reef taxa.

In this paper we focus on reef fishes because this group plays important functional roles in ecosystems, from herbivores which potentially control primary producer biomass (*Mumby, 2009*), to top predators that potentially control diverse mid trophic level taxa (*Glynn & Enochs, 2011*; *Holmlund & Hammer, 1999*). Moreover, fishes are economically relevant as they support commercial fisheries and recreational activities (sport fishing and scuba diving; *Leenhardt et al., 2015*). Given their role in shaping marine habitats and community structure, as well as their importance for human livelihoods, it is critical to evaluate if temporal changes in fish species in MUMPAs can alter ecological processes and ecosystem services (*Miller, Roxburgh & Shea, 2011*; *Mouillot et al., 2013*).

Limited information is available about temporal changes of fish communities within MUMPAs, and most of this has been based on traditional metrics (species richness, density and biomass; *Lester & Halpern, 2008*; *Sala & Giakoumi, 2018*). This approach is useful and descriptive, but omits critical information about a broader range of changes that may occur due to protection, especially those related with ecosystem functioning (*Coleman et al., 2015*). In order to effectively describe responses of fish assemblages to protection in terms of biological diversity and their effects in ecological processes, recent studies have included functional information of species that reveal their role in reef ecosystems (*Bates et al., 2014a*; *Coleman et al., 2015*). Assessing functional diversity in MPAs has allowed the detection of greater heterogeneity in responses at species level (direct effects) and subtle changes at the community level over time (indirect effects; *Coleman et al., 2015*; *Claudet, 2017*).

In the current study we aimed to evaluate how fish assemblages changed through a 13-year study period at PNZMAES, using traditional and functional metrics. We analyzed temporal changes in the biomass of commercial species to detect direct effects of protection, and also assessed if no-take and "buffer" zones followed similar trends of change or stability in traditional and functional metrics through the study period. We tested three hypotheses: (1) fish species and functional diversity in PNZMAES remain stable through time because the aim of the MUMPA is to conserve ecosystems at the initial state (not to restore them); (2) biomass of commercial species is maintained over time, and does not increase as a result of the moderate fishing pressure that persists in the MUMPA; (3) fish assemblages in no-take zones present higher diversity and more stability because of the total fishing ban in these areas (acting as reserves), compared with "buffer" zones that possess lower diversity and slight changes through time due moderate fishing pressure. This study also covers information gaps in the analysis of MPAs because it incorporates novel metrics (functional diversity) and includes a continuous timeline that allows a more comprehensive approach when assessing diversity changes over time. Moreover, widespread increases in human population, tourism and fishing activities around PNZMAES make this MUMPA an ideal test case for assessing the effects of partial protection in the face of mounting anthropogenic pressures, as well as the potential value of replicating this management strategy more widely, especially in developing countries.
## MATERIALS & METHODS

### Study locations and data sampling

PNZMAES is located at the southwestern GC and encompasses an area of 486 km$^2$ (24°43′00″ to 24°22′44″N, 110°26′58″ to 110°17′11″W). This MPA was decreed in 2007 and, even before the management plan was implemented in 2014 (*CONANP-SEMARNAT, 2014*), park managers and fishermen established three levels of use in the area (Fig. 1); no-take zones where fishing is strictly prohibited (∼1.4% of the MPA); and "buffer" zones divided into two categories: traditional use zones where fishing activities with hook and line, and sport fishing are allowed (∼4.4% of the MPA); and sustainable zones where, in addition to the activities in the traditional zones, aquaculture activities are also permitted (∼94.2% of the MPA). Industrial fisheries (including purse seining, long-lining and trawling) are prohibited in the entire MUMPA (*CONANP-SEMARNAT, 2014*). Tourism activity exhibited an increase of 5% in the number of visitors from 2002 to 2007 (from 20,231 to 21,379 visitors; *CONANP-SEMARNAT, 2014*), with numbers concentrated in the no-take area of "Los Islotes", which is dedicated to protect the breeding colony of the sea lion *Zalophus californiensis* (*Buchoul, 2016*). More than two surveillance trips were performed per day and a total of 84 penalties were registered from 2013 to 2017 (personal communication). Although fishing information is not available for PNZMAES, fisheries catch databases from La Paz CONAPESCA office reported an increase of 133% of the fish total catch from 2005 to 2016 (*CONAPESCA, 2016*).

From 2005 to 2017, eleven sites were monitored twice a year, in cold (January to June) and warm (August to November) seasons (Fig. 1). In each visited site, six to eight underwater visual censuses of 100 m$^2$ or 60 m$^2$ were conducted. Different transect areas were considered in the study period since the methodology changed following a park management decision in 2009. Field surveys were approved by the dirección del Parque Nacional exclusivamente la zona marina del Archipiélago de Espíritu Santo of the Comisión Nacional de Áreas Naturales Protegidas (PROMOBI/IGCBCS/003/2015, CONANP/PROMANP/MD/DRPBCPN/02/2016, FOO.DRPBCPN.IGCBS.PNZMAES.-245-17).

To better consolidate the number of replicates, we aggregated data if visual census were separated by less than 200 m, and were performed the same day and depth range. From an original 1,546 sampling units we obtained a total of 320 transects (average = 333 ± 73 m$^2$, median = 300 m$^2$; Electronic Supplementary Material ESM1), with 10% of the surveys undertaken before the inauguration of the MUMPA (2007), 65% after the protection but before the implementation of the management plan (2008–2014), and 25% after the activation of specific rules for fisheries in the MUMPA (ESM2, Table S1). For each transect, data on the number of species (species richness) were collected, and with this information we constructed species accumulation curves (randomization method) to analyze if we had a large enough sample to adequately characterize the species pool in the study period. Using ANOVA we compared if there were differences between transect areas (180, 200, 240, 300, 360, 400, 540 and 600 m$^2$), given the aggregation of samples, as well as the changes in methodology and number of samples per field campaign. Species

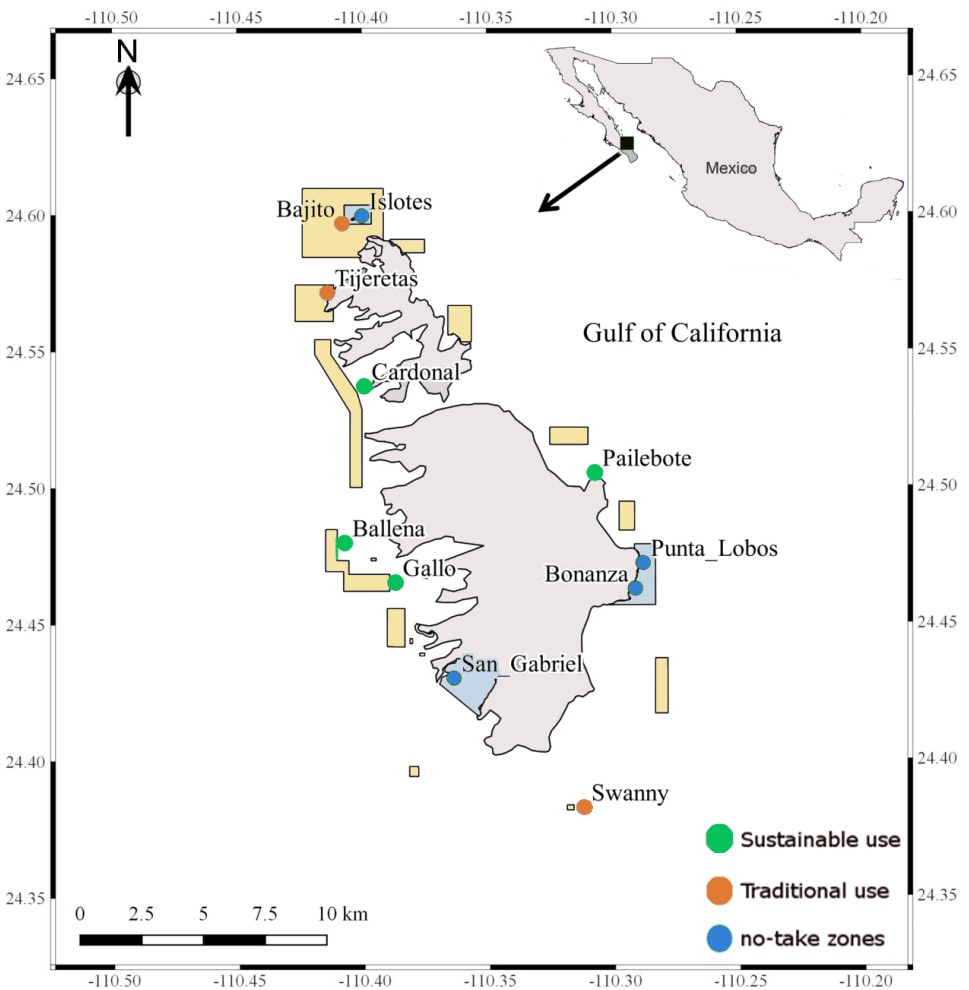

**Figure 1** **Location of the surveyed sites in the Parque Nacional exclusivamente la zona marina del Archipiélago de Espíritu Santo (PNZMAES).** The points represent each site surveyed and the levels of use are color-coded: No-take zones in blue (Islotes, Punta Lobos, Bonanza and San Gabriel), Traditional use zones in orange (Bajito, Tijeretas and Swanny), and Sustainable use zones in green (Cardonal, Ballena, Gallo and Pailebote). The rectangle-polygons represent the area with traditional use (yellow-filled) and the no-take zones (blue-filled).

accumulation curves and ANOVA were performed in R software with the "vegan" and "stats" R packages, respectively (*Oksanen et al., 2010*; *R Development Core Team, 2014*).

In addition, abundance and individual fish size (to the nearest 5 cm), were recorded for each transect. Modal size was estimated for fishes in schools. Fish biomass (g/m²) was estimated using the length-weight relation: *Weight  = a\*Length*^b, with coefficients *a* and *b* obtained from FishBase (*Froese & Pauly, 2009*).

## Biological traits of fishes and diversity indices

To estimate the functional diversity of the fish assemblages, each fish species observed was classified according to six categorical traits (nominal or ordinal) that reflect key

aspects of fish ecology (*Mouillot et al., 2014*): (1) maximum body size (ordinal: 0–7, 7–15, 15–30, 30–50, 50–80, >80 cm), (2) mobility (ordinal: low, medium within a reef, and high between reefs), (3) period of activity (nominal: diurnal and nocturnal), (4) gregariousness (ordinal: solitary, pairing, small groups and large groups), (5) vertical position in the water column (ordinal: benthic, bentho-pelagic and pelagic), and (6) diet (nominal: herbivores-detritivores, invertivores targeting sessile invertebrates, invertivores targeting mobile invertebrates, planktivores, apex predators targeting fish and cephalopods and omnivores; ESM1). The same categories have been used in a previous study in the GC (*Olivier et al., 2018*), and taken together they provide Functional Entities (FEs).

The FEs were used to build a categorical traits matrix that was transformed into a numerical matrix to calculate functional indices. Pairwise distances between species (according to their FE) were computed using the Gower dissimilarity; this coefficient was selected because it allows the use of different types of variables while giving them equal weight (*Gower, 1971*). A principal coordinate analysis (PCoA) was performed using this functional dissimilarity matrix. The number of axes used to calculate functional indices (first four axes) was determined *a posteriori* according to the method of *Maire et al.* (*2015*; Fig. 2). These PCoA scores were used to calculate three complementary functional diversity indices: functional richness, functional dispersion, and functional originality (*Villéger, Mason & Mouillot, 2008*; *Mouillot et al., 2013*). The definitions used here are the same as those of *Mouillot et al. (2013)* and *Mouillot et al. (2014)*. Functional richness was defined as the volume covered by a set of species (for example those observed in a transect), in proportion to the whole functional space encompassed by the outermost vertices of the assemblage, which is determined on the basis of the complete species suite (Fig. S1). This metric represents the range of functional niches found in a particular assemblage. Functional dispersion was defined as the weighted mean distance between species present in a sampling unit, and the weighted average position of the entire assemblage in the niche space (Fig. S1). This represents the functional complementarity between species. Functional originality was defined as the weighted mean distance between each species and its nearest-neighbor in the niche space (Fig. S1), thus, the opposite of functional redundancy. The functional dispersion and functional originality were weighted separately by the density and the biomass of each species. We used the function "dbFD" and "multidimFD" from the "FD" R package to calculate the different functional indices (*Laliberté, Legendre & Shipley, 2014*). Data exploration of the eight indices was carried out following the protocol described in *Zuur, Ieno & Elphick (2010)*. Density and biomass were log-transformed (base 2) to achieve normality and homoscedasticity.

## Statistical analyses

We ran linear mixed models (LMMs; *Bolker, 2015*) to compare the eight indices (species richness, density, biomass, functional richness, functional dispersion weighted by density and biomass, and functional originality weighted by density and biomass) against the three levels of use of the MPA, i.e., no-take, traditional and sustainable zones. These analyses included as random variables the potential effects of year, season and site; it is worth noting that by considering site as random variable, we could account for spatial

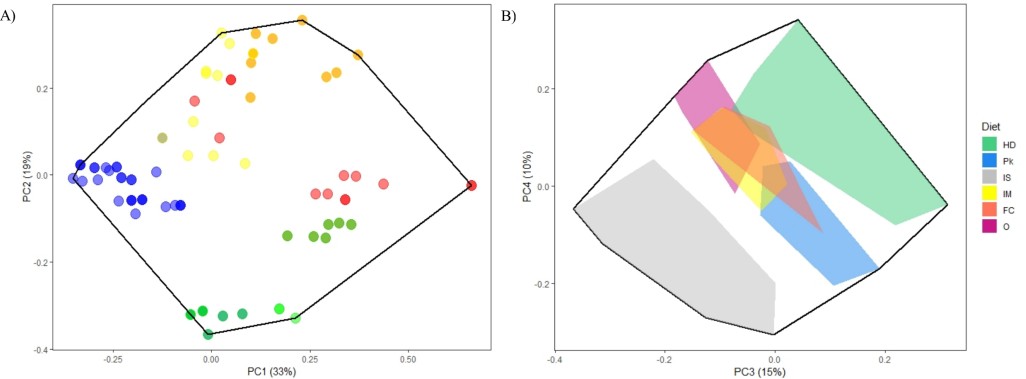

**Figure 2 Functional space (first four PCoA axes) of the fish species in PNZMAES.** (A) For PC1 and PC2, each point represents a species and the color corresponds to their functional traits: low mobility and gregariousness (blue), high mobility and gregariousness (red), bentho-pelagic or pelagic and diurnal species (yellow), benthic and nocturnal species (green). (B) The convex hulls for each trophic group are represented along PC3 and PC4: herbivores-detritivores (HD), planktivores (Pk), invertivores targeting sessile invertebrates (IS), invertivores targeting mobile invertebrates (IM), apex predators targeting fish and cephalopods (FC), and omnivores (O).

and temporal autocorrelation associated with repeated monitoring of the same sites (*Zuur et al., 2009*). Season (cold or warm) was also defined as a random variable as in the Gulf of California reef fish abundance and biomass both decrease when temperature is low (*Pérez-España, Galvan-Magaña & Abitia-Cárdenas, 1996*; *Aburto-Oropeza & Balart, 2001*; *Rodríguez-Romero et al., 2005*). Year was taken as a random variable because we wanted to compare the three levels of use independently of temporal changes, to determine if there were differences in the eight indices associated with the variations in fishing restriction.

In addition, LMMs also analyzed changes in fish diversity (eight indices considered) through the 13-year period. Again we considered the potential effects of season and site by including them as random factors, but in this case use level was not included in models since it did not present significant differences for most of the indices calculated in the previous analyses. Visual inspections of the residuals from each model did not reveal any severe violation of parametric assumptions.

To evaluate biomass changes in commercial species through the 13-year study period, we ran a LMM considering site and season as random factors. The species involved in this analysis were selected from a list of commercial fishes present at PNZMAES (*Niparajá, 2011*), although out of the 42 species established as target species for the MUMPA, only 28 were observed in our reef surveys and considered for the analysis (ESM1).

The final analysis involved identification of which common species (present in at least 50% of the transects; 26 species) either increased ("winners") or decreased ("losers") in occurrence (presence of species per transect), density and biomass, through the 13-year study period. To do so, we applied generalized linear mixed models (GLMMs) with a binomial distribution for occurrence, and with a negative-binomial distribution for density and biomass, to account for overdispersion in the residuals (*Zuur et al., 2009*). Density and biomass values for each common species were rounded to an integer, a

prerequisite to use the negative-binomial distribution, and site and season were considered as random variables. All statistical analyses were performed with the "lme4" R package (*Bates et al., 2014b*).

# RESULTS

Species accumulation curves (ESM2, Fig. S2A) indicate that 50 censuses provide an adequate sample to properly characterize the species pool in our area. Since we conducted a higher number of censuses in the field period, we have a standardized sample effort. Significant differences were found between transect areas ($F_{7,312} = 6.53$, $P = < 0.001$), particularly among 180 and 600 m$^2$ surveyed areas (ESM2, Fig. S2B), which represent less than 4% of the total sample size. Our principal transect areas (300 and 400 m$^2$, which represent 64% and 25% respectively), did not show significant differences ($P => 0.05$) between them.

The LMMs by level of use showed no significant differences ($P > 0.05$) in most of the indices except density ($P = 0.02$), which was higher where traditional fishing is allowed (Fig. 3; ESM2, Table S2). Temporal LMMs indicated that species richness significantly decreased by 13% on average (confidence interval (CI) = 19% to 5%) and biomass increased by 43% (CI: 2% to 99%) through the 13-year study period (Fig. 4; ESM2, Table S3). Nevertheless, the biomass of commercial species did not increase ($P = 0.30$; ESM2, Table S4). Functional richness decreased 24% through the study period (CI: 34% to 14%), while functional dispersion (complementarity among species), showed contrasting results: it increased 8% when weighted by density (CI: 6% to 11%), and decreased 13% when calculated with biomass (CI: 19% to 8%). No change was observed for density and functional originality (Fig. 4; ESM2, Table S3).

According to the LMMs (Fig. 4), the temporal trends were similar in the three use levels of the park when considering density and functional dispersion weighted by biomass (decrease), as well as biomass and functional dispersion weighted by density (increase). However, a biomass increase was more evident in the traditional use zones, and functional dispersion (calculated using fish density) was higher in the sustainable use zones (Figs. 4C and 4E). Concerning the species and functional richness, no significant changes were detected where traditional fisheries are permitted, although significant decreases were observed in the sustainable use and the no-take zones (Figs. 4A and 4D).

In relation to the study of changes in common reef fish species through time, we identified 26 fishes present in at least 50% of the 320 transects (Fig. 5). GLMMs indicated that seven of them decreased significantly in occurrence through time, and four of these "loser" species were located on the outer margins of functional space (Figs. 5D and 5G). Only two "winner" species increased significantly in occurrence (Fig. 5A; ESM2, Table S5), while 17 remained stable through time. On the other hand, 12 common species decreased significantly in density and they occupied a large part of the functional space (Figs. 5E and 5H), while six "winner" species were identified (Fig. 5B; ESM2, Table S6) and the remaining eight did not present significant changes. Last, in the case of biomass, the number of fish species that increased or decreased significantly was more balanced, with eight "winners" and six "losers" (Fig. 5C; ESM2, Table S7). The former were located near the aggregation

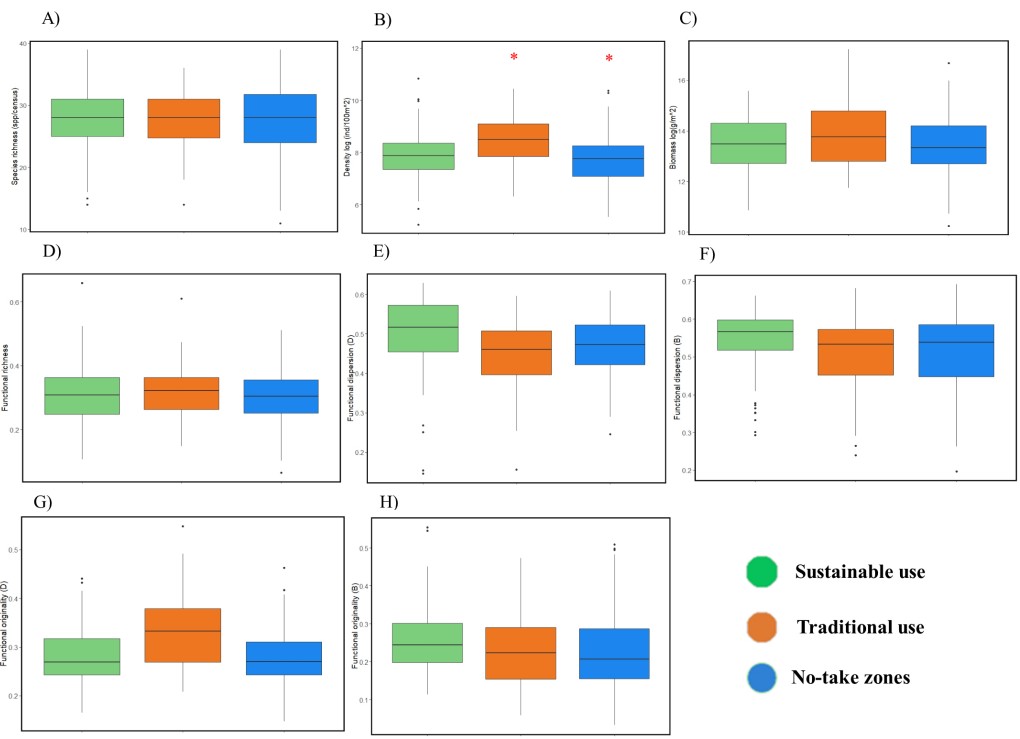

**Figure 3 Comparison of fish diversity between levels of use in PNZMAES by ANOVA ($\alpha = 0.05$).** The density and biomass have been log-transformed (base 2). The red asterisk indicates a significant difference among the levels of use. The boxplots depict the mean, 2nd and 3rd quartiles, the confidence interval (95%) and the outlier dots.

of functional centroids (expressed by the 95% ellipse), while the latter tended to occupy more outward positions (Figs. 5F and 5I). The other 12 species did not change in biomass through 13 years of monitoring.

## DISCUSSION

Considering that, by law, the aim of PNZMAES is to conserve ecosystems in a state similar to the original (before the establishment of the MUMPA; *CONANP-SEMARNAT, 2014*), MPA objectives would be met if no changes or increases in the values of species and functional indices are observed. That is precisely what we found: a temporal increase in biomass of the fish assemblage, and no significant changes in density and functional originality. However, our hypothesis of stability in fish diversity through time cannot be fully accepted, since some indices (species richness, functional richness and functional dispersion weighted by biomass) declined through the 13-year study period (Fig. 4; ESM2, Table S3).

Temporal analyses indicated that species richness declined through time (independently of the differences in sampling effort; ESM2, Fig. S2). This result can be linked to the decrease in functional richness, since some of the species that declined in the occurrence trend analysis ("losers") were located at the outer margins of functional space. Their

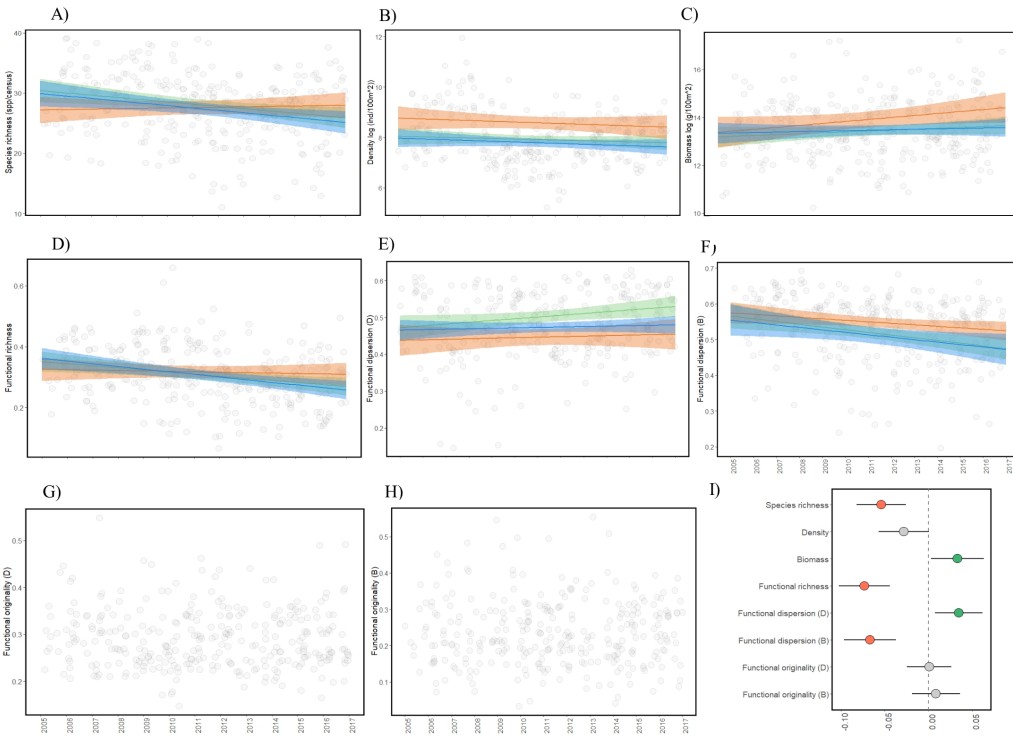

**Figure 4 Temporal trends of the fish diversity along a 13-year study period in PNZMAES.** Regression lines for linear models are shown for each ecological and functional index. A jitter position has been added to handle overplotting of the data. The colors of regression lines are according to the level of use in the park, i.e., green: sustainable, orange: traditional, and blue: no-take zones (A–H). The coefficient estimates (mean ± 95% confidence interval) of the LMMs considering the 11 sites are shown for the eight indices calculated (I). Coefficients have been standardized for visualization. Green and red circles show significant positive and negative changes, respectively. Grey symbols indicate no significant changes. D and B indicate if that index was weighted by density (D) or biomass (B).

disappearance or decrease in numbers caused a reduction in the functional space of the fish assemblage in these particular years (Fig. 5A; ESM2, Table S5). Fewer species and functional richness suggest local extinction of species may ultimately occur (some with unique functional role), thereby risking the maintenance of some ecosystem processes (*Mouillot et al., 2013*). A good example is the endemic GC damselfish *Chromis limbaughi*, which declined in presence and abundance in the park, and possesses an unusual role as a deep-water planktivore (*Robertson & Allen, 2015*).

Functional dispersion, representing the ecological complementarity between species, exhibited changes in the 13-year study period. Clearly the functional structure of the assemblage is dynamic, reflecting variability in the distribution of density and biomass among species (*Bates et al., 2014a*). Changes in functional dispersion were evident when calculated on the basis of density of individuals (Fig. 5B), a pattern associated with an increase in abundance of certain species ("winners") located far from the centroid (case 1 in Fig. S1), and a decrease in abundance of other species ("losers") distributed close to the centroid (case 2 in Fig. S1). This result demonstrates that in spite of the general stability
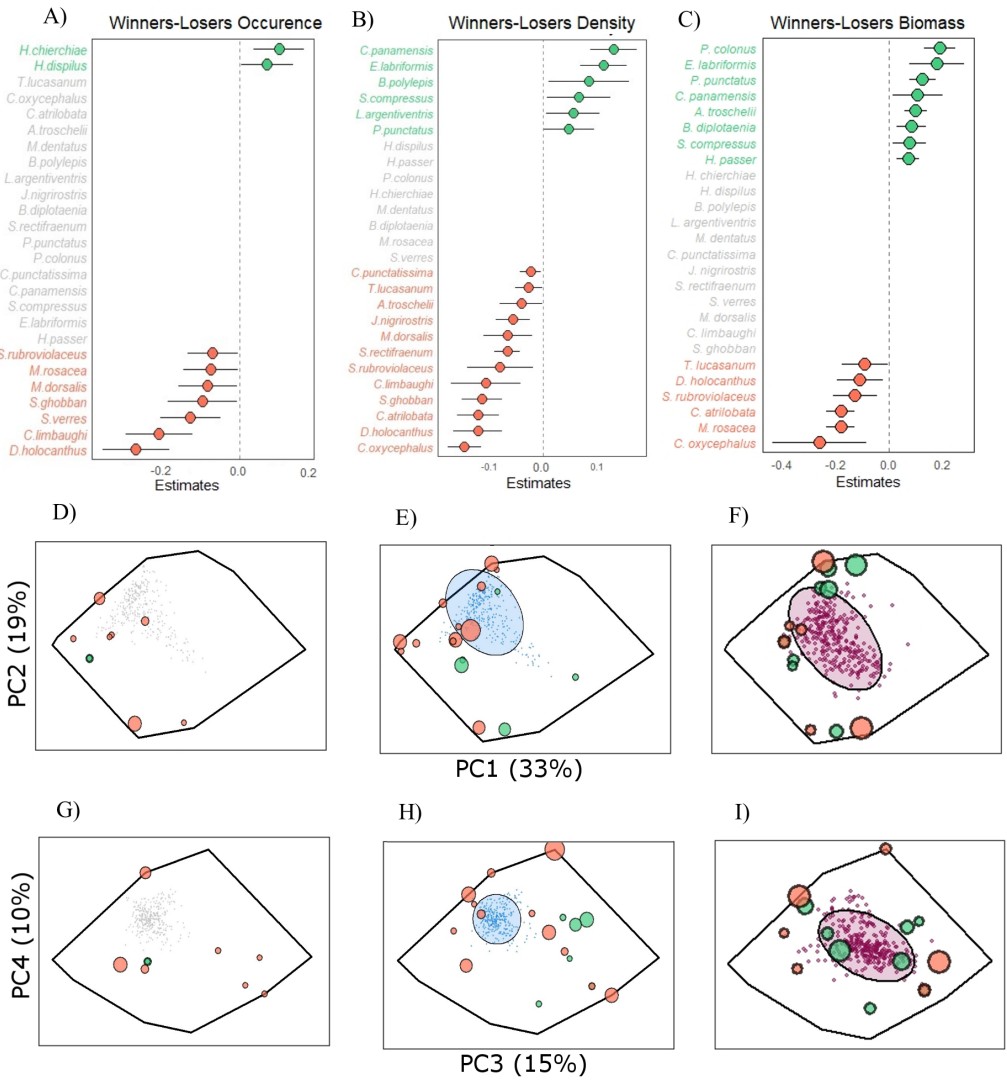

**Figure 5** **Winner and loser species over a 13-year study period in PNZMAES.** From (A) to (C), the winner and loser species in term of occurrence, density, and biomass. The mean coefficient estimates (±95% confidence interval) of the GLMMs show the effect of the year on the common species (present in at least 50% of the transects). Only the parameters of the species for which a significant effect was found, and for which the models were validated, are shown. Green and red circles indicate significant positive and negative values, respectively. The position in the functional space of winner and loser species is shown in the functional space for the occurrence (D, G), density (E, H), and biomass (F, I). The size of the dots is proportional to the $z$-values of the models. For the density and biomass functional spaces, the average-weight centroids (one per transect) are indicated by blue and purple dots, respectively. The 95% confidence interval ellipses of the centroids values are illustrated.

in density of fishes at PNZMAES (Fig. 4B; ESM2, Table S3 ) some functions are losing representation while others are gaining. This could be associated with effects of protection on species (i.e., predatory and competitive interactions) that have been observed elsewhere after an average of 13.1 ± 2.0 years from establishment of a MPA (*Micheli & Halpern, 2005*; *Babcock et al., 2010*). Moreover, the increase in abundance of species with different

functions (widespread "winner" species) could favor the resilience in the community by having more balanced, and thus similarly efficient complementary functions (*Mouillot et al., 2013*).

By contrast, functional dispersion calculated from biomass showed a decline through time, which could be explained by the distribution of biomass "winner" species around the center of the functional space, while the "loser" species tended to occupy more extreme positions (Figs. 5F and 5I). The fact that "winner" species were adjacent within functional space indicates that similar functions have been favored, mainly by the augmentation of size in their individuals, since density trend analysis did not show a similar pattern (Figs. 5E and 5H). Larger individuals of winner "species" have more impact in ecosystem functioning since their ecological rates (i.e., consumption and production) are higher compared with smaller individuals (*Cyr, Downing & Peters, 1997*), which could have caused changes in ecosystem processes in PNZMAES in the study period.

The rise in biomass despite the stability in density of fishes (Figs. 4B and 4C; ESM2, Table S3), can also be explained as the result of an increase in fish size (*Lester et al., 2009*). This was corroborated by analyzing the average size per year of the full fish assemblage, which revealed an increase of 10% between 2005 and 2017 (ESM2, Fig. S3B). Interestingly, in other MPAs this increase in average size is associated to direct effects of protection, in the form of return or growth of commercial species due the reduction of fishing pressure (generally apex predators with large size; *Leenhardt et al., 2015*).  However, at PNZMAES the biomass of commercial fishes did not exhibit significant changes through time (ESM2, Table S4), which implies that non-target species are responsible for this increase. Similar patterns were also observed in another GC MUMPA, Parque Nacional Bahía de Loreto, where biomass of herbivorous and planktivorous non-commercial species increased after 13-years of protection (1998–2010) in a small no-take area (1.27 km$^2$; *Rife et al., 2013*). Overall, this result corroborates our stability hypothesis for the commercially important species group due to the persistence of moderate fishing pressure in this MUMPA.

Predation and cascading trophic effects by commercial species do not seem to be the main drivers of change in fish assemblages in PNZMAES, as has been observed in other MPAs shortly after protection (5.13 ± 1.9 years; *Babcock et al., 2010*). Other factors, such as competitive interactions, habitat quality and fishing activities, could explain the simultaneous increase and decline of different species (*Leenhardt et al., 2015*). Possible examples of competition are the increase observed in the parrotfish species *Scarus compressus* and the decrease of *Scarus rubroviolaceus* (ESM1) in the density and biomass trend analyses (Tables S6 and S7, ESM2). Given that these species share all the ecological traits considered in this study (and thus could be considered redundant), the possible substitution among species should not represent a threat for their essential functions and ecosystem services in eastern Pacific reef environments. Such functions include bioerosion and later deposition of carbonate sediment, which helps in the construction of the reef framework, and the magnitude of herbivory, which influences nutrient cycles and controls algae proliferation (*Bellwood, Hoey & Hughes, 2011*).

Related to habitat quality, the particular case of a decrease in density and biomass of the coral hawkfish *Cirrithichthys oxycephalus* (Figs. 5B and 5C), is an example of a species
with unique function (ESM1) that is threatened under the conditions of PNZMAES. The decrease of this species has been linked to coral cover loss in Parque Nacional Cabo Pulmo, caused by repeated hurricane impacts and the 1997–1998 El Niño event (*Iglesias-Prieto, Reyes-Bonilla & Riosmena-Rodríguez, 2003*; *Alvarez-Filip & Reyes-Bonilla, 2006*). Considering that in PNZMAES coral cover significantly declined ($b = -0.44$; ESM2, Fig. S3A), habitat loss represents an important issue and focus for future conservation efforts in this MUMPA, to prevent local extinction of coral-associated species.

In the case of possible fishing effects, the decrease in occurrence and biomass of the commercially important leopard grouper *Mycteroperca rosacea* (Figs. 5A and 5C) is indicative of removal of large-sized individuals. Although IUCN has classified this species as "Least Concern" (*Erisman & Craig, 2018*), a decline in abundance has been reported from over a decade ago (*Sala et al., 2004*) because it is one of the most intensely fished groupers in the GC (*Sala et al., 2003*; *TinHan et al., 2014*). Given the key ecological role of the leopard grouper in reef ecosystems as a high level carnivore (in adult stages over 36.5 cm TL; *Moreno-Sanchez et al., 2019*), and its economic importance by representing a high proportion of the annual fishing revenue (78% in combination with snappers; *TinHan et al., 2014*), enhanced protection strategies are needed for stabilization and recovery of population numbers within the region.

Protection efforts for commercial species are, however, difficult because most of these species are large and have high movement capacity (*Munguia-Vega et al., 2018*), travelling distances (~15 km; *TinHan et al., 2014*) that can exceed the maximum span of local no-take zones (2.5 km). A balance between loss of fishes in "buffer" zones due fishing activities and emigration from no-take zones, could explain the observed functional homogeneity in space and time of PNZMAES fish assemblages, which presented similar values and temporal trends for most of the indices (Figs. 3 and 4). These results are not consistent with the hypothesis that ecological and functional performance is greater in no-take zones, because the entire MUMPA apparently behaves as a unit. In this particular case, an expansion of the no-take zones should help protect highly mobile commercial species (*Munguia-Vega et al., 2018*), and could clarify the benefits of a total fishing ban. Improved conservation outcomes have been reported in Parque Nacional Cabo Pulmo (25 km$^2$ of core area), which exhibited a 463% increase of total fish biomass and a 30% increase in biomass of predatory fishes between 1999 and 2009 (*Aburto-Oropeza et al., 2011*).

In addition to the expansion of the no-take areas, other local strategies such as surveillance of important zones and restrictions on use of fishing gears by local fishermen, should be enhanced to counteract the observed decline in abundance and occurrence of particular species that have deteriorated functional richness in the MUMPA (Figs. 4A and 4D). To conduct these programs, the collaboration of the local residents is essential if the conservation aim of the MPA is to be achieved. New developments, including the recent addition of PNZMAES to the IUCN Green List of Protected Areas (2018), provides an opportunity to obtain human and logistic resources to improve local management in co-responsibility with local stakeholders, and reverse the observed decline in fish diversity over the last decade.

In conclusion, this study demonstrates that the conservation aim of PNZMAES has not been fully accomplished because, despite a long-term increase in biomass and stability in density and functional originality, some indices significantly declined between 2005 and 2017. Declines in species and functional richness could indicate possible local extinctions and the loss of certain ecosystem processes, such as those associated with competitive interactions, habitat loss and fishing activities. However, changes in functional dispersion indicate that the functional structure of the fish assemblage is dynamic: while some species are losing representation through time, others are gaining. Furthermore, biomass increase was associated with a general increase in individual fish size through time, rather than to an increase in presence or growth of commercial species. Populations of commercial fishes remained stable in the face of moderate fishing pressure persisting in the area. Finally, no-take zones presented similar temporal trends and values of diversity in comparison with "buffer" zones, which indicates that the entire MUMPA is behaving as a unit, possibly through dispersal and connectivity between zones. Local management strategies, which should include expansion of the no-take areas, habitat preservation, fishing regulations and support of local community, should be taken to overturn the decline in particular species and to promote the conservation of fish functional diversity in this and other MUMPAs in the GC.

## ACKNOWLEDGEMENTS

We thank Guillaume Rieucau, James Junker and Charlotte Dromard for their helpful comments and suggestions to improve our manuscript, to CIBNOR for the facilities and support to develop this research, to Sociedad de Historia Natural Niparajá, A.C. and Comisión Nacional de Áreas Naturales Protegidas for provide the information, and to all the people and institutions (CICIMAR, UABCS and Niparajá) who participated in field surveys of the monitoring program of PNZMAES.

### Funding

Monitoring programs were funded by Comisión Nacional de Áreas Naturales Protegidas (PROMOBI/IGCBCS/003/2015 y CONANP/PROMANP/MB/DRPBCPN/02/2016), Sociedad de Historia Natural Niparajá, A. C., David & Lucile Packard Foundation, Sandler Family Foundation, The Walton Family Foundation, The Waterloo Foundation. Georgina Ramírez-Ortiz received a scholarship from CONACYT (266599) for her Doctorate degree. The funders had no role in study design, data collection and analysis, decision to publish, or preparation of the manuscript.

### Grant Disclosures

The following grant information was disclosed by the authors:
Comisión Nacional de Áreas Naturales Protegidas.
Sociedad de Historia Natural Niparajá, A. C., David & Lucile Packard Foundation.

David & Lucile Packard Foundation.
Sandler Family Foundation.
The Walton Family Foundation.
The Waterloo Foundation.
CONACYT: 266599.

## Competing Interests

The authors declare there are no competing interests.

## Author Contributions

- Georgina Ramírez-Ortiz conceived and designed the experiments, performed the experiments, analyzed the data, prepared figures and/or tables, authored or reviewed drafts of the paper, coordinated the group, and approved the final draft.
- Héctor Reyes-Bonilla conceived and designed the experiments, performed the experiments, analyzed the data, authored or reviewed drafts of the paper, funding, and approved the final draft.
- Eduardo F. Balart conceived and designed the experiments, performed the experiments, authored or reviewed drafts of the paper, funding, and approved the final draft.
- Damien Olivier conceived and designed the experiments, performed the experiments, analyzed the data, prepared figures and/or tables, authored or reviewed drafts of the paper, edited figures, and approved the final draft.
- Leonardo Huato-Soberanis conceived and designed the experiments, authored or reviewed drafts of the paper, answered to reviews, and approved the final draft.
- Fiorenza Micheli conceived and designed the experiments, authored or reviewed drafts of the paper, edited english, and approved the final draft.
- Graham J. Edgar conceived and designed the experiments, analyzed the data, authored or reviewed drafts of the paper, edited english, and approved the final draft.

## Field Study Permissions

The following information was supplied relating to field study approvals (i.e., approving body and any reference numbers):

Field surveys were approved by the dirección del Parque Nacional exclusivamente la zona marina del Archipiélago de Espíritu Santo of the Comisión Nacional de Áreas Naturales Protegidas (2015: PROMOBI/IGCBCS/003/2015; 2016: CONANP/PROMAN-P/MD/DRPBCPN/02/2016; 2017: FOO.DRPBCPN.IGCBS.PNZMAES.-245-17). Before 2014 the authorities at the National Park only required a notice that field surveys were performed.

## Data Availability

The raw data is available in the Supplemental Files.

## Supplemental Information

Supplemental information for this article can be found online at http://dx.doi.org/10.7717/peerj.8885#supplemental-information.

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
