# Peer review of "Reduced fish diversity despite increased fish biomass in a Gulf of California Marine Protected Area"

_PeerJ, doi:10.7717/peerj.8885_

## Round 0.1 · original submission · Major Revisions

Dear Dr. Ramirez-Ortiz,

I have now received the evaluation of two reviewers on your manuscript entitled "Reduced fish diversity despite increased fish biomass in a Gulf of California marine protected area". Both recognized that your work has potential, but they recommend that you make substantial amendements to you manuscript to reach the scientific standard of publication. In particular, one of the reviewers was very critical about how you discussed the results reported in your study. I recommend that you take a particular care to address this criticism as I echo the reviewer's opinion.

All the best and thank you for considering PeerJ to publish your work.

Dr. Guillaume Rieucau

·

Basic reporting

Basic reporting:
English usage: English usage could use minor revision. For example, there are instances of disagreement between sentence subject and verbs (e.g., line 91: “…this group play important…”, line 284: “ CONAPESCA inspectors’ controls…”).
References/Background: Functional metrics are conceptually underdeveloped in the paper. The differences in inference and conclusions drawn between ‘traditional’ ecological metrics and functional metrics needs more background in the introduction and more discussion in discussion/conclusions. In its current form, functional metrics are introduced in the last two sentences of the introduction (line 97) and are only introduced to say they have recently been used. Besides the area of functional metrics, literature cited is adequate.
Structure/Figures/Data: The article structure is acceptable. Raw data provided is easy enough to navigate and appears robust. Figures are clean and well done. Many of the figure and table captions need to be reworked, as they inadequately describe the figure/table and lack necessary information to allow figure/table to be assessed independently. For example, the caption for Figure 1 needs to include that points on the map relate to the sampling locations (Also, PNZMAES versus PNAES in the figure). Further suggestions are included within the attached files.
Self-contained: The current manuscript needs to further motivate the questions and hypotheses/predictions for the study. As it is currently written, there are no or weakly developed questions and reasoning for the use of functional metrics. As such, it is unclear whether the data presented, and response variables chosen, address the questions or were chosen because they have been used in other publications. If correctly reframed in revisions, data would likely be adequate.

Experimental design

Experimental design
Aims & Scope: The research fits the scope of the journal.
Research question: Not clearly defined research question/s and hypotheses/prediction. The objective as stated is, “In the current study we aimed to evaluate how fish assemblages changed through a 13-year study period at PNZMAES”. This does not motivate the question about areas of different use, for example. See above and attached file for further comments.
Rigor/Methods: The methods, as written, lack the details to fully assess whether certain aspects of the study can be addressed. I highlight a few I find very important: 1) there is inadequate information about the underwater visual censuses to have an idea of the precision, bias, and power to detect changes. Were the censuses conducted by a single diver? Was there any attempt (e.g., training, repeat censuses, etc.) to estimate inter-diver bias amongst the, likely, many divers over the decade of sampling? 2) Looking at the raw data, there is a general decrease in sampling effort from the beginning of the sampling period to the last years (~400 m2 to 300 m2). Can the decrease in species richness be explained by the decrease in effort? Where are the researchers on the yield-effort curve? What effort is needed to adequately characterize the species pools? Is this something to be concerned or have they accounted for this? 3) Line 105-106: Does the study represent a 3- or 13-year period of “MPA effect”? This is important for how the study is framed and how much inference can be drawn about the impacts of a management action versus ‘natural’ ‘background’ patterns/trends. This should be made very explicit in the methods and in framing the study.

Validity of the findings

Validity of findings
Data: See above.
Conclusions: The discussion needs a large revision. There is inadequate discussion around the results of the manuscript in the last half of the discussion. Further, there are areas of speculation that need to be noted as such and sections that move beyond speculation, beyond tangential to the questions, and near the point of unsubstantiated accusations against an international non-profit that are not suited for this forum.

Additional comments

General overview: Ramírez-Ortiz and colleagues explore temporal trends in ecological and functional diversity of reef fish assemblages of a multi-use marine protected area within the Gulf of California. They claim a loss of species and certain functional diversity metrics and a lack of enhancement of commercial fish stock bring in to question whether the goals of this MPA are being met. The data the authors present is interesting and provides for the comparison of fish assemblage responses to management with relatively long-term data.
However, I cannot recommend publication of the manuscript in its current form and suggest that substantial revisions would be necessary. I have 5 general areas that I would like to highlight as especially important to address, and also provide more specific and line by line comments in attached documents. Important areas for improvement/revision are:
1) The introduction doesn’t clearly state questions and hypotheses/predictions or motivate the use of functional metrics over ‘traditional’ ecological metrics. Why are functional metrics useful, how can they “…sensitively detect early changes in assemblage structure and instability, through redundancy and complementary processes”? and can predictions be made about how management of MPAs should impact these metrics? Is increasing functional dispersion (for example) a predicted positive outcome of no-take zones?
2) The methodology surrounding the UVCs was inadequate to assess if errors, precision, biases were accounted for through the decade long study. This is especially important because there seems to be a decrease in the effort (transect area surveyed) through time. If conclusions about species richness are made, there needs to be assurances that richness measurements are not dependent upon effort. I admit this is slightly outside of my expertise, but if 300 m2 is well established to be adequate to characterize species richness in reefs and richness estimates are well saturated at this amount of effort a relevant citation would suffice. If this is not established, the authors should be able to show that richness measurements are well saturated and not dependent on effort so as to allow comparisons through time.
3) It needs to be clearly stated how long management has been implemented in the park if conclusions about how management/MPA use zones are to be made. Are there 2-3 years of pre-MPA data (2005-2007) or ~9 years (2005-2014)? This is needs to be clear for readers to assess the possible implications for the trends observed.
4) It seems like there is a clear result of shifting body size distribution towards larger body size individuals in the assemblage over time. This appears to lead to divergent and opposing temporal patterns in ecological and functional metrics between density and biomass. I suggest this is an important trend that cannot be dismissed, as loss of density may be, partially, attributable to body size shifts to larger species and individuals within a species. The current lack of comment on it anywhere in the manuscript is notable.
5) The discussion needs a large revision, especially the last half. There is a large ‘digression’ that distracts from the findings in the paper and is not suited for this forum. Instead, more attention should be paid towards the results in an objective manner. Much of the discussion focuses on negative trends or losses of function. But depending on whether density or biomass is looked at the opposite observations can be made in many cases. Right now, it seems like there is a bias towards highlighting shortcomings. If explicit questions and hypotheses and predictions were clearly stated earlier, the reporting of the results and choice of what to highlight would seem less arbitrary.

·

Basic reporting

This short manuscript, untitled "Reduced fish diversity despite increased fish bioass in a Gulf of California marine protected area" presents interesting results due to the long-term database (13 years).
This study reaches the scope and aims of the revue but need corrections before publication.
My concerns are about the presentaton of the results, that are sometimes difficult to read (figures) or in supplementatry material.

Title : add a upper-case letter to "marine protected area"

L30-31 : « we detected a reduction in fish biodiversity… » In which zone ? This result should be detailed.

L32-33 : a part of this sentence is missing.

L34-35 : reword this sentence.

L55 : « it is rapidly incrasing », reword this sentence. For exampe, by « it is currently growing rapidly »

L80-88 : I found that information on anthropogenic impacts/influence or human activities in the MPA are missing in the introduction ?
Frequentation of the MPA, numbers of tourists per year ? In which area ? The quantity of fished organisms in the sustainable and artisanal areas ? As well as the effort of management in the MPA (number of monitor guards ? penalities to people that do not respect the rules ?)

Literature references are well cited and listed at the end of the manuscript.
Only one change to do :
L272 : change « Bellwood et al. 2011 » by « Bellwood, Hoey & Hughes, 2011 »

A lot of tables (statistical results) are given in supplementary materiel, that I think is a shame.

The raw data are shared and are clearly presented.

Experimental design

The methods is well presented and well isllustrated with scientific references.

Figure 1 : where are located the sustainable use areas ? Does the MPA encompasses the islands ? The limits of the MPA and the different zones are not clear in Figure 1.

L120 : « we obtained 320 transects… ». it could be usefull to have the numbers of transects done per year (before or after the establishment of the MPA) and per area (no-take, sustainable, traditional).

L139-149 : the definitions of the three indices and the mode of calculation are not clear, and it is difficult to understand the meaning of these indices based on Figure 2.

Figure 3 : authors should add in the caption, the meaning of « (B) » and « (D) » in the legends axes.

Figure 3 : the caption of figure 3 should be « Fish diversity, density and biomass according to the levels of use in the PNZMAES » (same remark for Figure 4, this figure shows also the evolution of fish density and biomass, not only diversity).

Validity of the findings

Figure 3 : fish density is the only parameter that showed difference between the three zones areas of the MPA. Do densities are different between the three zones ou between « traditional use zone » and the two other ? It looks like black and blue boxplot are not different.

Figure 4 : The red and green circles are refering to which zone area ? It appears that specific richness increases in traditional use area, while a red circle is mentionned in the small graph.

Figure 4 : grey lines (sustainable use) are not visible.

L184 : « species richness and density decreased while biomass increased.. » That means that mean fish size increased ? Why the fish sizes are not presented in the different figures ? Which species increase in size ?

L305 : isn’t there some information about benthic composition of the reef during the survey census period ? Evolution of coral cover or macroalgae cover ?

L339 : « habitat alteration, lowering in larval recruitment and global change » can also be included in the anthropogenix impacts (not as « natural factors »).

---

## Round 0.2 · accepted · Accept

Dear Dr. Ramírez-Ortiz,

I am pleased to inform you that after receiving the report of one reviewer and my careful read of your manuscript, your article has been accepted for publication in Peer J. Both the Reviewer and I recognized that this version is much more improved and addresses now all the comments previously made during the review process.

Congratulations!
Best regards,
Guillaume Rieucau
Academic Editor, PeerJ

·

Basic reporting

The language of the text has been much improved. A few minor edits include:
Line 60: 'persist' to persists
Line 64: add 'to' following due...
Line 144: subtle changes at *the* community level
Line 191: 'of' not if
Line 364: 'increase' not increment

Experimental design

Revised manuscript clearly defines questions, addresses my concerns regarding changing effort surveys throughout the study, and greatly improves the description of what was done.

Validity of the findings

Fine.

Additional comments

Authors should be commended for their efforts to address my previous concerns in this revised article. It is much improved in my opinion.